# Factors associated with undernutrition among pregnant women in Haramaya district, Eastern Ethiopia: A community-based study

**Meseret Belete Fite**[1]*, **Abera Kenay Tura**[2,3], **Tesfaye Assebe Yadeta**[2], **Lemessa Oljira**[4], **Kedir Teji Roba**[2]

**1** Department of Public Health, Institute of Health Sciences, Wollega University, Nekemte, Ethiopia, **2** School of Nursing and Midwifery, College of Health and Medical Sciences, Haramaya University, Harar, Ethiopia, **3** Department of Obstetrics and Gynaecology, University Medical Centre Groningen, University of Groningen, Groningen, The Netherlands, **4** School of Public Health, College of Health and Medical Sciences, Haramaya University, Harar, Ethiopia

* meseretphd2014@gmail.com

**Data Availability Statement:** All relevant data are within the manuscript.

**Funding:** This study was fully funded by Haramaya University after the proposal has been defended.

## Abstract

### Introduction

Although undernutrition in pregnancy has continued to get global attention as pregnancy is considered a critical period in the life cycle owed to increase the metabolic and physiological demands, evidence is scarce on undernutrition and associated factors among pregnant women in eastern Ethiopia. Therefore, this study assessed the undernutrition and associated factors among pregnant women in Haramaya district, Eastern Ethiopia.

### Methods

A community-based cross-sectional study was conducted among randomly selected pregnant women in Haramaya district, eastern Ethiopia. Data were collected through face-to-face interviews, anthropometric measurement, and hemoglobin analysis by trained research assistants. An adjusted Prevalence ratio (aPR), and a 95% confidence interval (CI), were used to report associations. Poisson regression analysis model with a robust variance estimate identified variables associated with undernutrition. Data were double entered using Epi-data 3.1 and cleaned, coded, checked for missing and outliers, and analyzed using Stata 14 (College Station, Texas 77845 USA. Finally, the p-value <0.05 was the cut-off point for the significant association.

### Results

A total of 448 pregnant women with a mean age of 25.68 (± 5.16) were included in the study. The prevalence of undernutrition among pregnant women was 47.9% (95% CI: 43%-53%). From the analysis, the undernutrition was more likely higher among respondents who had five or more family members (APR = 1.19; 95% CI = 1.02–1.40), lower dietary diversity (APR = 1.58; 95% CI = 1.13–2.21) and those who were anemic (APR = 4.27; 95% CI = 3.17–5.76).

The funder has no role in conception, design of the study, statistical analysis, result interpretation and in writing up the manuscript. The funding institution has no role in the publication consent or approval.

**Competing interests:** The authors have declared that no competing interests exist.

## Conclusion

Nearly half of the pregnant women in study area were undernourished. High prevalence was found among women who had large family sizes, low dietary diversity and anemia during pregnancy. Improving dietary diversity, strengthening family planning services and giving special attention to pregnant women, supplementation of iron and folic acid, and early detection and treatment of anemia is essential to improve the high burden of undernutrition and the adverse effect on pregnant women and the fetus.

## Introduction

Undernutrition refers to deficiency primarily of calories, and overall inadequate consumption of food and nutrients to provide an individual's requirement to support good health [1]. Moreover, undernutrition occurred due to the double burden of increased demands during pregnancy and inadequate intake of food during pregnancy [2]. Undernutrition is a key contributor to maternal mortality and morbidity, and adverse birth outcomes [3]. Mid-upper arm circumference *(MUAC)* is a proper measure for screening undernutrition during pregnancy [4]. MUAC is a good indicator of the protein reserves of a body, and a thinner arm reflects wasted lean mass and most appropriate anthropometric measure to detect short-term changes in th*e* nutritional status [5].

Worldwide, nearly about 462 million pregnant women had malnutrition [6]. In low-resource countries, undernutrition among pregnant women is continuing to increase unremarked, as the main predictor of adverse birth outcomes [7, 8]. The reports of studies indicate that vulnerability to undernutrition in utero is linked with impaired growth and development in childhood, short stature in adults, reduced academic achievement and decreased economic productivity [9, 10]. Although literature points to the association of maternal undernutrition with adverse birth outcomes, little is documented about the risk predictors that influence prenatal nutritional status. Prenatal undernutrition is unacceptably high in developing countries [11], and Africa is the utmost severely overwhelmed [12]. More than one-fifth of Ethiopian women are exposed to malnutrition during their pregnancy [13] and, the risk is 68% higher among rural women compared to urban women [14].

Although Ethiopia has made a striding change in maternal health death over the last decades, undernutrition during pregnancy remains a significant public health issue with prevalence ranging from 14.4% in Gonder [15] to 44.7% in Gumay district [16]. Several studies indicated that factors including, maternal age, residency, literacy, marriage before 18 years old, ANC follow-up, meal frequency, meal skipping, and household food security [16–20], were associated with maternal undernutrition. However, these studies have documented that the level of magnitude undernutrition and associated risk factors among pregnant women vary across the agro-ecological setups [19].

Although the Ethiopian ministry of health has tried to implement health extension program strategies to reduce maternal undernutrition, studies indicate malnutrition among pregnant women persistently remains a serious public health problem in the country [15–20]. Moreover, evidence is scarce on undernutrition and associated factors among pregnant women in the Haramaya district. Therefore, this study assessed the undernutrition and associated factors among pregnant women in Haramaya District, Eastern Ethiopia.

## Methods

### Description of the study area

As a detailed description has been given elsewhere in the previous paper [21], the study was embedded into the Haramaya Health Demographic Surveillance and Health Research Centre

(HDS-HRC), established in 2018. The HDS-HRC covers 12 rural kebeles (the lowest administrative unit in Ethiopia) out of 33 found in the district located approximately 500 KM from the capital city, Addis Ababa. Of 5252 pregnant women in the district during the study period, 2306 were followed by the HDS-HRC [22]. This study was conducted from January 5 to February 12, 2021

### Study design and period

A community-based cross-sectional study was conducted from January 5 to February 12, 2021.

### Source population and study population

All pregnant women living in the district constituted the source population; whereas all pregnant women who lived in the selected kebeles for at least six months during the study period were the study population.

### Inclusion and exclusion criteria

Participants were a part of pregnancy surveillance initiated in HDS-HRC. For the reason that dietary practice is affected by the local social and cultural values, all pregnant women who lived a minimum of six months in the district were involved in this study. However, all pregnant women with reported acute and chronic illnesses, seriously ill and unable to communicate during the study period were excluded

### Sample size determination and sampling procedures

The sample size was determined using single and double population proportion formulas with their corresponding assumption, and the largest sample size was considered. As such, the sample was computed using the single population proportion formula with the following assumptions: 95% confidence interval, the prevalence of undernutrition among pregnant women Gumay District, (44.9%) (16), 5% marginal error, and 10% non-response rate; the final computed sample size was 419. However, since this study was part of a larger longitudinal study (a prospective cohort study aimed to assess neonates' birth weight and the association with maternal iron status), the same 475 pregnant women were included. A detailed description has been given elsewhere in the previous papers [21, 23, 24].

### Data collection and measurement

Data were collected through face-to-face interviews, anthropometric measurement, and serum ferritin analysis by trained research assistants. The questionnaire contained data on socio-economic, obstetric, maternal perception, food consumption, dietary diversity, knowledge, attitude, and practices of pregnant women. In addition, mid-upper arm circumference (MUAC) and maternal height measurements were taken. The nutritional status of the pregnant women was measured with non-stretchable MUAC tape and the reading value was taken to the nearest 0.1-cm. All measurements were performed threefold and the average value of two concordant readings was considered as the ultimate value. Pregnant women with average MUAC measurements of less than 23 cm were categorized as having "undernutrition" otherwise normal [25, 26]. The questionnaire was initially prepared in English and translated to the local language (Afan Oromo) by individuals with good command of both languages. It was also pre-tested on 10% of the samples in Kersa District before actual implementation. Women's hemoglobin concentration (in g/dL) was measured at each study site by well-trained medical technologists using Hemo-Cue® Hb 301 system, according to the manufacturer's instructions (HemoCue AB Ängelholm

Sweden) which is a gold standard for fieldwork. A prick was done on the tip of the middle finger after the site was cleaned with disinfectant. The first drop of blood was cleaned off and the second drop was collected to fill the microcuvette which is then placed in the cuvette holder of the device for measuring hemoglobin concentration. Hemoglobin values were adjusted for altitude as per the Center for Disease Prevention and Control (CDC) recommendation [27].

As the detailed description has been given elsewhere in a previous papers [23, 24], the formerly validated food frequency questionnaire (FFQ) containing 27 of the most common lists of food items consumed by the district community was used to assess the dietary diversity of the study participants [28–33]. The food items in the FFQ were grouped into ten food groups, including cereal, white roots and tubers, pulse and legumes, nuts and seeds, dark green leafy vegetables, other vitamin A-rich fruits and vegetables, meat, fish and poultry, dairy and dairy product, egg, other vegetables, and other fruits. The sum of each food group pregnant women consumed over seven days was calculated to analyze the dietary diversity scores (DDS) [32]. Furthermore, the dietary diversity score was converted into tertiles, with the highest tertile labeled as a "high dietary diversity score" whereas both lower tertiles combined were defined as a "low dietary diversity score". The food variety score (FVS) is the frequency of individual food items consumed during the reference period. Therefore, it was estimated by calculating each individual's intake of the 27 food items over seven days.

## Data quality assurance

Two training days were given for data collectors, laboratory professionals, and supervisors before the pre-test. The questionnaire pre-test was conducted on 10% of the sampled pregnant women in a district that was not included in the main study; appropriate adjustments were made based on the results. Supervisors closely managed data collection, checking the data daily before entry. The investigators administered all data collection activities. In addition, laboratory analysis quality assurance was maintained and trained and experienced laboratory professionals strictly followed standard operating procedures for all parameters.

## Data processing and analysis

Data were double entered using Epi-data 3.1. Data were cleaned, coded, checked for missing and outliers, and analyzed using Stata 14 (College Station, Texas 77845 USA). Frequencies, percentages, summary measures and tables were used to describe and present the descriptive information of respondents. The MUAC is a much simpler anthropometric measure than the BMI, as its use eliminates the need for expensive equipment, such as height charts and scales, and the need for calculations. It is also much easier to perform on a patient who is acutely unwell, bed bound or sedentary. Another important advantage of using MUAC is that there is minimal change in the MUAC during pregnancy, so it may be a better indicator of pre-pregnancy body fat and nutrition than the BMI. The outcome variable (undernutrition) was dichotomized as undernutrition (coded as 1) and normal (coded as 0). Poisson regression analysis models with a robust variance estimate were fitted to identify predictors of undernutrition. Next, the binary analysis variables with a $p < 0.25$ were entered into the adjusted log-binomial models. Results were presented using the crude prevalence ratio (CPR) and adjusted prevalence ratio (aPR). Akaike's information criterion (AIC) and Bayesian information criterion (BIC) were used to test for model fitness. The goodness-of-fit was assessed using the Pearson chi-square and deviance tests, with the statistical significance level at alpha = 5%. The explanatory variables were examined for multi-collinearity before taking them into the multivariable model using a correlation matrix for the regression coefficients, the standard errors, and the variance inflation factor value.

The wealth index was employed to estimate the economic level of families. The wealth dispersion was generated by applying the principal component analysis (PCA). The index was calculated based on the ownership of latrines, agricultural land and size, selected household assets, livestock quantities, and source of drinking water, a total of 41 household variables. The previous paper [28] described nutritional knowledge and attitudes toward consumption of an iron-rich diet using the Likert scale applying the PCA; the factor scores were totaled and classified into tertiles. Women's autonomy was evaluated using seven validated questions adopted from the Ethiopian Demographic Health Survey [34]. For each question, the response was coded as "one" when the decision was made by the woman alone or jointly with her husband, or "zero" otherwise. The detailed description has been given elsewhere in a previous papers [21, 23, 24].

## Ethical consideration

This study was conducted in agreement with the Declaration of Helsinki-Ethical principle for medical research involving human subjects [35]. The proposal was approved by the Institutional Health Research Ethics Review Committee (IHRERC) of the College of Health and Medical Sciences, Haramaya University (ref No: IHRERC/266/2020). Written informed consent was obtained from all participants and legally authorized representatives "of minors below 16 years of age and illiterates," and confidentiality was maintained by excluding all personal identifiers

## Operational definition

**Undernutrition.**   Nutritional status of pregnant women measured by MUAC was labeled as under-nutrition when

MUAC<23 cm, otherwise normal [25].

**Anemia.**   Anemia was defined as a Hemoglobin level of < 11.0 g/dl during the first or third trimester or <10.5 g/dl during the second [36].

**Mid-upper arm circumference (MUAC).**   Is used as a measure of fat-free mass and a measurement of the circumference of the upper arm at the midpoint between the olecranon and acromion processes [26].

**Nutritional knowledge.**   Was measured through16 nutritional knowledge questions on the feature of nutrition needed in pregnancy and the score was computed by conducting PCA. Then composite was ranked into tertiles [29].

**Educational status.**   Respondents who had grade at least grade one education level were labeled as"formal", whereas respondents those could able to read or write sentences were categorized as" Informal education"

## Results

### Socio-demographic characteristics

Out of 475 eligible pregnant women, the study included 448, yielding a 94.3% response rate. The mean age of the women was 25.68 (±5.16), ranging from 16 to 36. The majority of the respondents could not read or write (73.88%), were housewives (96.1%), farmers (93%), and had a family size of 1–5 (76.56%). Only 20.09% were in the wealthiest quintiles (Table 1).

### Anthropometric and nutritional status of respondents

Among 448 respondents, 47.9% (95% CI: 43%- 53%) were undernourished and 45.98% were anemic. Of the total respondents, 29.46%, 37.50%, 24.8%, and 26.12% of them had high dietary diversity, high food variety score, high consumption of ASFs, and > 4 meal frequency respectively, Table 2.

**Table 1. Socio-demographic of pregnant women in Haramaya district, Eastern Ethiopia, 2021 (n = 448).**

| Variables | Frequency(n) | Percentage (%) |
|---|---|---|
| Age (years) | | |
| <18 | 25 | 5.58 |
| 18–35 | 400 | 89.29 |
| >35 | 23 | 5.13 |
| Mean (± SD) | 25.68 (± 5.16) | |
| Educational level of the woman | | |
| Can't read or write | 331 | 73.88 |
| Read or write | 26 | 5.81 |
| Formal education | 91 | 20.31 |
| Educational level of husband | | 49(23.33) |
| Can't read or write | 259 | 57.81 |
| Read or write | 61 | 13.62 |
| Grade 1–8 | 102 | 22.77 |
| Grade 9 and above | 26 | 5.8 |
| Occupation of the woman | | |
| Housewives | 433 | 96.65 |
| Merchants | 15 | 3.65 |
| Occupation of husband | | |
| Farmers | 420 | 93.75 |
| Daily labors | 28 | 6.25 |
| Family size | | |
| 1–5 | 343 | 76.56 |
| ≥5 | 105 | 23.44 |
| Agricultural land possession | | |
| No | 271 | 60.49 |
| Yes | 177 | 39.51 |
| Wealth Index (Quintile) | | |
| Poorest | 90 | 20.09 |
| Poor | 90 | 20.09 |
| Middle | 89 | 19.87 |
| Rich | 90 | 20.09 |
| Richest | 89 | 19.87 |

## Factors associated with undernutrition

In the bi-variable analysis, women's educational level, stage of pregnancy, family size, dietary diversity, consumption of ASFs, skipping meals, anemia status, antenatal care, perceived confidence, food restriction, and khat chewing were found to be a candidate for multivariable analysis at p<0.25. Using the Poisson regression analysis model with a robust variance estimate, undernutrition was more likely higher among respondents who had more than five family members (APR = 1.19; 95% CI = 1.02–1.40), low dietary diversity (APR = 1.58; 95% CI = 1.13–2.21) and had anemia during pregnancy (APR = 4.27; 95% CI = 3.17–5.76), Table 3.

## Discussion

Despite the encouraging improvement in maternal death in, undernutrition among pregnant women remains a public health issue in in Ethiopia [34]. In this study, we reported the prevalence of the undernutrition and associated factors among pregnant women in Haramaya

**Table 2. Anthropometric and nutritional status of pregnant women in Haramaya district, eastern Ethiopia, 2021 (n = 448).**

| Variables | Frequency(n) | Percentage (%) |
|---|---|---|
| Nutritional status | | |
| Normal | 233 | 52 |
| Undernutrition | 215 | 48 |
| Anemia status | | |
| Anemic | 206 | 45.98 |
| Non-anemic | 242 | 54.02 |
| Dietary diversity | | |
| Low | 316 | 70.54 |
| High | 132 | 29.46 |
| Consumption of ASFs | | |
| Low | 337 | 75.22 |
| High | 111 | 24.78 |
| Food Variety Sore (FVS) | | |
| Low | 280 | 62.50 |
| High | 168 | 37.50 |
| Meal frequency | | |
| < 4 | 331 | 73.88 |
| ≥ 4 | 117 | 26.12 |

district. We found that the prevalence of undernutrition among study participants was 47.9% (95% CI: 43%-53%) and was noted to be nearly half. Moreover, the risk factors of undernutrition were higher among women who had greater than five family sizes, low dietary diversity and were anemic.

The present finding is comparably consistent with studies conducted in Northwest Tigray, Ethiopia [37] and South West Ethiopia [16]. However, the result of current is higher than studies carried out in Gambela, Ethiopia [19] and Eastern Ethiopia [13] and southern Ethiopia [38], Kenya [39], Sudan [40], and Nigeria [41]. The possible variation might be due to culturally diverse countries, thus it is not important to introduce direct correlation of the current the result with the findings of the studies carried out in different countries. Farmers in the Haramaya district have been growing khat for many years and are a major cash crop in this study area. Moreover, khat chewing amongst pregnant women is common in the district [22]. Therefore, the higher prevalence of undernutrition in this study setup might be due to increased nutritional demands in pregnancy and the decrement of dietary intake as of the effects amphetamine found in khat to reduced appetite [42]. On other hand, the difference in methods and measures used might contribute to the variations. In our study we used a community-based cross-sectional study design to assess the undernutrition and associated factors s among pregnant, whereas, some previous of the studies were carried out at the institutional level.

In the present study, we observed that having low dietary diversity was independently associated with maternal undernutrition during pregnancy, which is in agreement with studies conducted in different parts of Ethiopia [19, 43, 44]. The inappropriate dietary practice among pregnant women was noted in this study [23]. This is might be due to maternal dietary habits, food taboos, and cultural beliefs that can affect nutrition during pregnancy and women do not consume additional meals during the pregnancy. Nutritious diets, essential nutrition services and optimal nutrition practices are essential to prevent all forms of malnutrition before and

**Table 3. Factors associated with undernutrition among pregnant women in Eastern Ethiopia, 2021.**

| Variables | Undernutrition | | CPR(95%CI) | APR (95%CI) | P-value |
|---|---|---|---|---|---|
| | Yes | No | | | |
| | (n = 215) | (n = 233) | | | |
| **Educational level of women** | | | | | |
| Can't read or write/Informal | 177(82.33) | 180(77.25) | 1 | 1 | 0.334 |
| Formal | 38(17.67) | 53(22.75) | 0.73 (0.46,1.16) | 1.12(0.89,1.41) | |
| **Stage of pregnancy** | | | | | |
| First trimester | 8(3.72) | 11(4.72) | 1 | 1 | 0.755 |
| Second trimester | 137(63.72) | 159(68.24) | 1.18 (0.46,3.03) | 0.89 (0.74,1.30) | |
| Third trimester | 70(32.56) | 63(27.04) | 1.53 (0.58,4.04) | 0.96 (0.72,1.27) | |
| **Family sizes** | | | | | |
| 1–5 | 152(70.70) | 191(81.97) | 1 | 1 | 0.028* |
| ≥5 | 63(29.30) | 42(18.03) | 1.88 (1.21,2.94) | 1.19 (1.02,1.40) | |
| **Dietary diversity** | | | | | |
| High | 32(14.88) | 100(42.92) | 1 | 1 | |
| Low | 183(85.12) | 133(57.08) | 0.23 (0.147,0.37) | 1.58 (1.13,2.21) | 0.008* |
| Consumption of ASFs | | | | | |
| low | 182(84.65) | 155(66.52) | 1 | 1 | 0.777 |
| High | 33(15.35) | 78(33.48) | 2.39 (1.74, 3.28) | 1.05 (0.75,1.46) | |
| **Skipping meals** | | | | | |
| No | 79(36.74) | 83(35.62) | 1 | 1 | 0.784 |
| Yes | 18(63.26) | 150(64.38) | 0.95 (0.65,1.40) | 1.03 (0.85,1.24) | |
| Anemia status | | | | | |
| Non-anemic | 42(19.53) | 200(85.84) | 1 | 1 | < 0.001** |
| Anemic | 173(80.4) | 133 (14.16) | 4.84 (3.656.41) | 4.27 (3.17,5.76) | |
| Antenatal care | | | | | |
| No | 69(32.09) | 95(40.77) | 1 | 1 | 0.111 |
| Yes | 146(67.91) | 138(59.23) | 1.46(0.99,2.15) | 1.14(0.97,1.33) | |
| **Perceived confidence** | | | | | |
| No | 168(78.14) | 168 (72.10) | 1 | 1 | 0.318 |
| Yes | 47(21.86) | 65 (27.90) | 0.72(0.47,1.11) | 0.90 (0.73,1.11) | |
| **Food restriction** | | | | | |
| No | 133(61.86) | 166 (71.24) | 1 | 1 | 0.807 |
| Yes | 82(38.14) | 67 (28.76) | 1.53(1.03,2.27) | 1.02(0.85,1.23) | |
| **Khat chewing** | | | | | |
| No | 73(33.95) | 107 (45.92) | 1 | 1 | 0.261 |
| Yes | 142(66.05) | 126 (54.08) | 1.65 (1.13,2.42) | 1.10(0.93,1.31) | |

CPR = Crude Prevalence Ratio; APR = Adjusted Prevalence Ratio, CI = Confidence Interval at 95%

APR, CI and P-Value were found from multivariable Poisson regression analysis model with a robust variance estimate

** Statistically significant at p-value <0.001

* Statistically significant at p-value <0.05

during pregnancy. Therefore, nutrition education and counseling of pregnant women is critical for every antenatal care and should be intensified.

Anemia during pregnancy has maternal and perinatal various effects and it increase the risk of maternal and perinatal mortality [45, 46]. This study observed that pregnant women with anemia were more likely to be undernourished. The proportion of undernutrition was

significantly more among anemic pregnant women compared to normal hemoglobin level pregnant women. This result is comparably in agreement with studies employed in Walayita Sodo town, Southern Ethiopia [47], in Gonder northwest Ethiopia [15], and India [48], which shows the risk of undernutrition tends to increase among anemic women. This could be due to the reality that anemic pregnant women have a greater risk of being inadequate in micronutrients and therefore more likely to be undernourished. The nether reason might be the fact that khat chewing is the most common in this study area and most of the women chew khat, which could decrease.

Having a large family size was one of the determinants, which were independently associated with undernutrition during pregnancy. This study revealed that pregnant women who were from greater than five members of households were a greater prevalence of undernutrition, which is in line with studies conducted in different parts of Ethiopia [16, 19] and Western Nepal [20]. The result could be because, food insecurity is more common in households with large family sizes, women play a sacrificial role and are more vulnerable to being undernourished than other family members [49]. Large family sizes may lead to inadequate food intake. In Ethiopian culture, women habitually served their meals after all family members are addressed. Thus, pregnant women are more exposed to food insecurity and associated with inadequate nutrient intakes for two fundamental reasons. First, the physiological changes occur during pregnancy. Women's nutrient needs increase during pregnancy and lactation. Maternal nutrient needs increase during pregnancy and breastfeeding, and when these needs are not met, it may contribute to wasting and fatigue. Second, women have a sociological vulnerability. Studies reveal that, during periods of decreased food supply, women expose to reduced consumption comparative to men. Furthermore, women are expected to decline their consumption to safe those of babies and small children [50].

Trained health workers and medical laboratory technologists collected and analyzed the socio-demographic data and blood samples. One strength of this study is the use of hemoglobin as an indicator of nutritional status, which is preferable to a community-based study. Various limitations need to be considered when interpreting our results. Since the study was cross-sectional, limiting the causal inference between under-nutrition and its correlates.

## Conclusion

This study finding has shown that the prevalence of that undernutrition among pregnant women in Haramaya district is high. In addition, dietary diversity, family size, and anemia in pregnancy were identified as factors that hindered their maternal status. Therefore, it is important that nutrition education and counseling are given during each antenatal visit should be intensified. Nutritional counseling and intervention should be tailored to meet the need of pregnant women and to improve their dietary practice and good nourishment. We suggest nutrition policy, programs and interventions should be aimed at encouraging prenatal dietary practice focusing on dietary guidance, and raising awareness on the benefit of quality diet in pregnancy for both the mother and the newborn. Strengthening family planning services and giving special attention to pregnant women, supplementation of iron and folic acid, and early detection and treatment of anemia are suggested.

## Acknowledgments

Special thanks go to the Haramaya district health office staff for their enormous support during the data collection period. Finally, we like to thank all the women who participated in the study, the data collectors, and the supervisors.

## Author Contributions

**Conceptualization:** Meseret Belete Fite, Abera Kenay Tura, Tesfaye Assebe Yadeta, Lemessa Oljira, Kedir Teji Roba.

**Data curation:** Meseret Belete Fite, Abera Kenay Tura, Tesfaye Assebe Yadeta, Lemessa Oljira, Kedir Teji Roba.

**Formal analysis:** Meseret Belete Fite, Abera Kenay Tura, Tesfaye Assebe Yadeta, Lemessa Oljira, Kedir Teji Roba.

**Funding acquisition:** Meseret Belete Fite, Abera Kenay Tura, Tesfaye Assebe Yadeta, Lemessa Oljira, Kedir Teji Roba.

**Investigation:** Meseret Belete Fite, Abera Kenay Tura, Tesfaye Assebe Yadeta, Lemessa Oljira, Kedir Teji Roba.

**Methodology:** Meseret Belete Fite, Abera Kenay Tura, Tesfaye Assebe Yadeta, Lemessa Oljira, Kedir Teji Roba.

**Project administration:** Meseret Belete Fite, Abera Kenay Tura, Tesfaye Assebe Yadeta, Lemessa Oljira, Kedir Teji Roba.

**Resources:** Meseret Belete Fite, Abera Kenay Tura, Tesfaye Assebe Yadeta, Lemessa Oljira, Kedir Teji Roba.

**Software:** Meseret Belete Fite, Abera Kenay Tura, Tesfaye Assebe Yadeta, Lemessa Oljira, Kedir Teji Roba.

**Supervision:** Meseret Belete Fite, Abera Kenay Tura, Tesfaye Assebe Yadeta, Lemessa Oljira, Kedir Teji Roba.

**Validation:** Meseret Belete Fite, Abera Kenay Tura, Tesfaye Assebe Yadeta, Lemessa Oljira, Kedir Teji Roba.

**Visualization:** Meseret Belete Fite, Abera Kenay Tura, Tesfaye Assebe Yadeta, Lemessa Oljira, Kedir Teji Roba.

**Writing – original draft:** Meseret Belete Fite, Abera Kenay Tura, Tesfaye Assebe Yadeta, Lemessa Oljira, Kedir Teji Roba.

**Writing – review & editing:** Meseret Belete Fite, Abera Kenay Tura, Tesfaye Assebe Yadeta, Lemessa Oljira, Kedir Teji Roba.

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
