## [Decision Letter · Decision Letter 0]

18 Mar 2022

PONE-D-21-17941Determinants of Under-nutrition Among Pregnant Women in Haramaya District, Eastern EthiopiaPLOS ONE

Dear Dr. Fite,

Thank you for submitting your manuscript to PLOS ONE. After careful consideration, we feel that it has merit but does not fully meet PLOS ONE’s publication criteria as it currently stands. Therefore, we invite you to submit a revised version of the manuscript that addresses the points raised during the review process.

The manuscript has been evaluated by two reviewers, and their comments are available below.

The reviewers have raised a number of major concerns. They request improvements to the reporting of methodological aspects of the study and more information on how the data collection was completed. The reviewers also note concerns about the statistical analyses presented and request re-analyses be completed.

The reviewers suggest certain references be added to the text. While you are welcome to do so if you feel they are good suggestions you are not obligated to do so.

Could you please carefully revise the manuscript to address all comments raised?

We look forward to receiving your revised manuscript.

Kind regards,

Thomas Phillips, PhD

Staff Editor

PLOS ONE

Journal Requirements:

2. Please include additional information regarding the survey or questionnaire used in the study and ensure that you have provided sufficient details that others could replicate the analyses. For instance, if you developed a questionnaire as part of this study and it is not under a copyright more restrictive than CC-BY, please include a copy, in both the original language and English, as Supporting Information. If the original language is written in non-Latin characters, for example Amharic, Chinese, or Korean, please use a file format that ensures these characters are visible.

3. Please state whether you validated the questionnaire prior to testing on study participants. Please provide details regarding the validation group within the methods section.

The authors would like express sincere appreciation to Haramaya University for funding of this study. 

This study was fully funded by Haramaya University after the proposal has been defended. The funder has no role in conception, design of the study, statistical analysis, result interpretation and in writing up the manuscript. The funding institution has no role in the publication consent or approval.   

5. Thank you for submitting the above manuscript to PLOS ONE. During our internal evaluation of the manuscript, we found significant text overlap between your submission and the following previously published works, some of which you are an author.

- https://www.hindawi.com/journals/aph/2018/1350195/

- https://www.dovepress.com/metabolic-syndrome-among-working-adults-in-eastern-ethiopia-peer-reviewed-fulltext-article-DMSO

- https://journals.plos.org/plosone/article?id=10.1371/journal.pone.0247085

- https://link.springer.com/article/10.1186/s40748-018-0087-z

Please revise the manuscript to rephrase the duplicated text, cite your sources, and provide details as to how the current manuscript advances on previous work. Please note that further consideration is dependent on the submission of a manuscript that addresses these concerns about the overlap in text with published work.

Reviewers' comments:

Reviewer's Responses to Questions

**Comments to the Author**

1. Is the manuscript technically sound, and do the data support the conclusions?

Reviewer #1: Yes

Reviewer #2: Partly

2. Has the statistical analysis been performed appropriately and rigorously? 

Reviewer #1: Yes

Reviewer #2: No

3. Have the authors made all data underlying the findings in their manuscript fully available?

Reviewer #1: Yes

Reviewer #2: No

4. Is the manuscript presented in an intelligible fashion and written in standard English?

Reviewer #1: No

Reviewer #2: No

5. Review Comments to the Author

Reviewer #1: Abstract

Under Methods

Line 6: Authors need to add the word cross to Sectional study

Line 8: Authors need to change Data “was” to were

Line 9: Authors need to change was to were after food frequency questionnaires

Introduction

Paragraph one

Authors need to get more updated statistics since the figure of 3.5 million is from a 2008 reference.

Consider citing (Sserwanja, Q, Kawuki, J, Mutisya, LM, et al. Underweight and associated factors among lactating women in Uganda: Evidence from the Uganda demographic health survey 2016. Health Sci Rep. 2021; 4:e356. https://doi.org/10.1002/hsr2.356) with updated statistics

Paragraph four

Line 2: Authors need to remove s between and & nutritional status

Methods

Paragraph 3: Source population and study population

Authors need to revise the grammar of the first sentence

Paragraph 4: Inclusion and exclusion criteria

Authors need to revise the grammar of the first and second sentences

Paragraph 6

Authors need to change data was to data were in the first sentence

Authors need to elaborate to the readers why they choose MUAC as a method of assessing undernutrition and why not other methods

Paragraph 8: Analysis

Authors need to clarify if by normal being coded as 0, did this include overweight and obese women, or they were unable to measure this?

Were descriptives done? If yes, authors need to mention this.

RESULTS

Paragraph 1:

Authors need to explain why they chose to have cannot read and write, informal and formal education as the sub-categories. What was the cut off for formal education? Was a primary school candidate the same as a tertiary institute candidate? How was informal education defined?

Why was the classification of husband education different from that of the woman?

Under husband education, how different was can read and write from high school and above?

Note: Authors need to describe how these were agreed upon in the methods section

Gravidity and Parity are the almost same. Why did the authors add both?

Discussion

Paragraph 3:

Revise grammar of first sentence

Conclusion

Revise grammar of second sentence

References

Reference 8: Change to (Sserwanja, Q., Mukunya, D., Habumugisha, T. et al. Factors associated with undernutrition among 20 to 49 year old women in Uganda: a secondary analysis of the Uganda demographic health survey 2016. BMC Public Health 20, 1644 (2020). https://doi.org/10.1186/s12889-020-09775-2). It is the same paper but the one cited was a thesis and this is the published article

Kindly revise all the other references to ensure that they are complete with journal name, issue/volume and page number (OR as per journal’s requirement)

Reviewer #2: Review Reports to the authors

Manuscript Id: PONE-D-21-17941

Title: Determinants of Under-nutrition Among Pregnant Women in Haramaya District, Eastern Ethiopia

Comments

*While the title and the objectives talk about determinants of under-nutrition, the authors used a cross sectional study and reported only factors associated with under-nutrition, not the determinants. The authors should have used case control study. You have to modify the title as “Factors associated with under-nutrition among Pregnant Women in Haramaya District, eastern Ethiopia”, or otherwise the evidence presented doesn’t match your title and objectives.

*I suggest to write “Among” as ‘among’ and “Eastern” as ‘eastern’ in your revised title

You used different font styles for similar headings and sub-headings. Please check and revise it.

*The manuscript is full of grammatical and editorial errors. Please go through it line by line (thoroughly) and revise it, better checked by language professionals. Some of such errors are indicated below.

Abstract

Methods: - sectional study design was….

*But, what is sectional study design? Please make it clear.

- Data was collected with interviewers-administered questionnaires by well-trained health professionals.

*Make it data were collected with…. And this is also same under method section.

*well trained health professionals….who are these, HDSS staff? What do you mean by well trained? Please clearly indicate what their profession is, their level of education is and how their training was for this data collection. Describe these details under your methods section.

Results: line 2- “Under-nutrition” is written in different font from others.

-Line 3 Breakfast should be written as ‘breakfast’

Conclusion: -The the present study“

*What is “the the”? Please edit it.

- High prevalence “was observed on” women who “reported chat less than four times”

* was observed on should be written as “was found among”

*reported chat less than four times should be “ reported to chew Khat less than four times”

-Nutrition policy, programs and interventions….

* This is too general and ambiguous conclusion. Based on your findings specify to the point and indicate which policy, program and intervention should do what to improve which problem.

Introduction

*This section is not well written since doesn’t indicate the picture of maternal under-nutrition during pregnancy from global, regional, national and local contexts.

* The literatures are not well searched, and hence the introduction is too shallow. So many other related articles were missed.

*What is the importance of this study since the factors identified were already well known as you indicated under paragraph three? What is special now in this study? Can’t we use studies from other settings in Ethiopia for eastern Ethiopian regions?

-Line 2 ….affecting the healthy of women both in developed and developing countries, where, more than…

* write it as ‘affecting health of women both in developed and developing countries where more than…

Line 3…Moreover, under-nutrition occurred due to the…

*Write it as ‘Moreover, under-nutrition occurs due to the….

-Paragraph 4 line 2 ….and s nutritional status among

* re-write it as … and nutritional status among

Methods

*Chat has to be written as ‘Khat’, revise it accordingly throughout the document

*You excluded all pregnant women with reported acute and chronic illnesses. How trustable this reported illness is? And what number or how many percent of the women were excluded with these criteria?

-The sample size required and adequate for estimating the determinants of dietary practices of pregnant women was computed….

*Are you calculating for determinants of dietary practice or under nutrition? Be consistent, clear and to the point in line with your objectives.

*For sample size determination you just stated of using single and double population proportions. Would you indicate the double proportion formula and calculations used? *How many variables were used to try calculating your sample size using double proportion formula? Please indicate these as well.

* You described the role of agro-ecolocolgical difference on under-nutrition, but used study from Gumay district for sample size determination. Why? Any effect it might pose?

* Also mention names of those kebeles (with their total sample allocated), you used under the DSS and techniques followed using a simple diagrammatic presentation.

-After translation, it’s the consistency * write it as “its consistency”

* Indicate/cite literatures you used for developing your questionnaire

- Hemoglobin analysis was carried out in the health post located in each Kebele by laboratory technologists.

* How was this practical? You were collecting data home to home but hemoglobin at health center? Were you calling each women to health center? What is the need to go to health center since you used Hemocue?

-Hemoglobin level was adjusted for altitude before the data were entered.

*How was the adjustment done- any formula? What about adjustment for other factors- did you considered any? Indicate how you classified anemia under analysis part.

* standard operating procedures (SOPs)….indicate these under the procedures….only SOP you used.

* How was anthropometric measurement standardized? Any TEM (technical error of measurements) done? Try to answer this in relation to the data collectors training, profession and how were they trained as raised above?

* under analysis- you still stated “to identify determinants of under-nutrition”….you rather should say factors associated with under-nutrition, not determinants…..And you may also need to re-consider your method of analysis appropriate to this…..logistic regression is one.

* What about those women under age regarding consent to participate? You had 24 women under the age of 18 years.

Results

*How do you see 100% response rate in a community based study like yours?

*Table 1- would you classify and present women’s education similar to husbands’ case?

*How do you see the wealth index vs women’s and husband’s occupation? Mismatch?

* Table 2: nutritional status ….write it as under-nutrition and normal….check typos errors in the table and also do not mix acute…make it consistent from title to conclusion.

* Chat chewing….make it “Khat”

* Table 3…factors determinants of under-nutrition…..vs the sub-title factors associated with??? And also vs manuscript title, check it again. I suggest this analysis method to be changed as indicated above.

* And also try to present some of your data with other data presentation techniques than table…

Discussion

*This section is very shallow. You are not supposed just to say similarity and difference ,rather explain what the public health implication of each finding is and interprete.

E.g. You put “This might be the gio-ecological variation of study set up, methodology used, where some of them were conducted at facility level which does not really show the real magnitude of the problem as of community level”.

*You have to tell the readers how geo-ecological variation matters, what method difference brought difference in the findings,,,,etc?

*Re-look at the strengths and limitations you put…a lot of limitations with less strengths.

Conclusion

*How would you comment on wealth status and high under-nutrition in your findings?

*Check comments to the comments to conclusion under abstract

* And re-look at your recommendations regarding anemia and Iron folic acid supplementation.

Consent for publication

-Not applicable

*Authors should agree and give their consent for the publication of this paper…check that.

Reference

*Check other references are exhaustively searched and included. Reference style also needs some editorial corrections in line with journal requirements.

The end!

6. PLOS authors have the option to publish the peer review history of their article (what does this mean?). If published, this will include your full peer review and any attached files.

Reviewer #1: **Yes: **Sserwanja Quraish

Reviewer #2: No

---

## [Author Response · Author response to Decision Letter 0]

26 Jul 2022

Dear Editor! we have tried to revise the comments . Since this is the part of large longitudinal study from which some papers were recently published the methods section share some charactestics and overlap. However We have tried to refere with citation . Others are edited according to your request .

---

## [Decision Letter · Decision Letter 1]

10 Oct 2022

PONE-D-21-17941R1Factors associated with undernutrition among pregnant women in Haramaya District, Eastern Ethiopia: A community-based studyPLOS ONE

Dear Dr. Fite,

Thank you for submitting your manuscript to PLOS ONE. After careful consideration, we feel that it has merit but does not fully meet PLOS ONE’s publication criteria as it currently stands. Therefore, we invite you to submit a revised version of the manuscript that addresses the points raised during the review process.

We look forward to receiving your revised manuscript.

Kind regards,

Jianhong Zhou

Staff Editor

PLOS ONE

Journal Requirements:

Reviewers' comments:

Reviewer's Responses to Questions

**Comments to the Author**

1. If the authors have adequately addressed your comments raised in a previous round of review and you feel that this manuscript is now acceptable for publication, you may indicate that here to bypass the “Comments to the Author” section, enter your conflict of interest statement in the “Confidential to Editor” section, and submit your "Accept" recommendation.

Reviewer #1: (No Response)

2. Is the manuscript technically sound, and do the data support the conclusions?

Reviewer #1: Yes

3. Has the statistical analysis been performed appropriately and rigorously? 

Reviewer #1: Yes

4. Have the authors made all data underlying the findings in their manuscript fully available?

Reviewer #1: Yes

5. Is the manuscript presented in an intelligible fashion and written in standard English?

Reviewer #1: Yes

6. Review Comments to the Author

Reviewer #1: Thank you so much for working on the comments.

General comments

Kindly line number the manuscript and in the responses, add pages and line numbers for easier follow up

In the methods, kindly add a section describing the independent variables and how they were classified

Specific comments

Kindly see my comments on your responses:

Your response

Reviewer#1 comment: Authors need to elaborate to the readers why they choose MUAC as a method of assessing undernutrition and why not other methods

Authors’ response: Thank you so much. We appreciated your comments. We have clearly described why we use MUAC as a method of assessing under-nutrition in the revised manuscript. We have stated as the following

“The body mass index (BMI, kg/m2) is currently the gold standard for measuring body fatness. However, pregnancy-associated weight gain and oedema, as well as late booking into antenatal care in our population setting, causes us to question the reliability of using the BMI to assess body fat or nutritional status in pregnancy. The MUAC is a much simpler anthropometric measure than the BMI, as its use eliminates the need for expensive equipment, such as height charts and scales, and the need for calculations. It is also much easier to perform on a patient who is acutely unwell, bed bound or sedentary. Another important advantage of using MUAC is that there is minimal change in the MUAC during pregnancy, so it may be a better indicator of pre-pregnancy body fat and nutrition than the BMI”

My comment:

I seem not to see this. Kindly line show which pages this information is.

Your response

Reviewer#1 comment: Were descriptives done? If yes, authors need to mention this.

Authors’ response: Thank you so much. We appreciated your comments. We have to do descriptive analysis of socio-demographic characteristics and anthropometric and nutritional status of respondents

My comment

Kindly add this in the data processing and analysis section

Your response

Authors need to explain why they chose to have cannot read and write, informal and formal education as the sub-categories.

What was the cut off for formal education? Was a primary school candidate the same as a tertiary institute candidate? How was informal education defined?Why was the classification of husband education different from that of the woman? Under husband education, how different was can read and write from high school and above?

Authors’ response: Thank you a lot for your substantial effort. We have revised the women’s educational sub-categories as “cannot read and write, can read and write and formal education “. We had collected all the educational level of the respondent. However most of the respondents cannot read and write. Thus we put sub-categories into three. Formal education is defined as the educational status with minim of grade one in school. Those women who had adult functional education were label as “can read and write”

Moreover we had collected all the educational level of the husbands. However since the information on their educational status was majorly distributed with in sub-categories of “cannot read and write, can read and write 1-8 Grade 1-8 and Grade 9 and above”. Thus we put the available information within sub-categories

My comment:

1. Did you base on the national guidelines to say grade 1 is formal education for women?

2. Why did you remove formal education sub-category for women and only considered can read and write and can not read and write in Table 3?

3. Kindly add this categorization in the methods section

Your response

Reviewer#1 comment:

Note: Authors need to describe how these were agreed upon in the methods section

Gravidity and Parity are the almost same. Why did the authors add both?

Authors’ response: Thank you so much. We appreciated your comments. We have used the stage of pregnancy in the revised manuscript. :

My comment

Its good to look at the stage of pregnancy but also good to maintain parity/number of children/gravidity. Any particular reason why you decided to remove this?

7. PLOS authors have the option to publish the peer review history of their article (what does this mean?). If published, this will include your full peer review and any attached files.

Reviewer #1: **Yes: **sserwanja Quraish

---

## [Decision Letter · Decision Letter 2]

21 Feb 2023

Factors associated with undernutrition among pregnant women in Haramaya District, Eastern Ethiopia: A community-based study

PONE-D-21-17941R2

Dear Meseret Belete Fite,

We’re pleased to inform you that your manuscript has been judged scientifically suitable for publication and will be formally accepted for publication once it meets all outstanding technical requirements.

Kind regards,

Fernanda Penido Matozinhos, Ph.D

Academic Editor

PLOS ONE

Additional Editor Comments (optional):

Dear Meseret Belete Fite,

Thank you for the opportunity to review this manuscript. I am grateful for the invitation.

After careful consideration, I feel the manuscript explores a very important topic to maternal and child health. The modifications made the manuscript come to a satisfying result.

Kind regards,

Reviewers' comments:

Reviewer's Responses to Questions

**Comments to the Author**

1. If the authors have adequately addressed your comments raised in a previous round of review and you feel that this manuscript is now acceptable for publication, you may indicate that here to bypass the “Comments to the Author” section, enter your conflict of interest statement in the “Confidential to Editor” section, and submit your "Accept" recommendation.

Reviewer #1: All comments have been addressed

Reviewer #3: All comments have been addressed

2. Is the manuscript technically sound, and do the data support the conclusions?

Reviewer #1: Yes

Reviewer #3: Yes

3. Has the statistical analysis been performed appropriately and rigorously? 

Reviewer #1: Yes

Reviewer #3: Yes

4. Have the authors made all data underlying the findings in their manuscript fully available?

Reviewer #1: Yes

Reviewer #3: No

5. Is the manuscript presented in an intelligible fashion and written in standard English?

Reviewer #1: Yes

Reviewer #3: Yes

6. Review Comments to the Author

Reviewer #1: (No Response)

Reviewer #3: I would like to congratulate the authors for their work and for looking carefully at the comments made in the previous round of review. The article brings important contributions to maternal and child health.

7. PLOS authors have the option to publish the peer review history of their article (what does this mean?). If published, this will include your full peer review and any attached files.

Reviewer #1: **Yes: **Quraish Sserwanja

Reviewer #3: No

---

## [Editor Report · Acceptance letter]

28 Feb 2023

PONE-D-21-17941R2 

Factors associated with undernutrition among pregnant women in Haramaya District, Eastern Ethiopia: A community-based study 

Dear Dr. Fite:

I'm pleased to inform you that your manuscript has been deemed suitable for publication in PLOS ONE. Congratulations! Your manuscript is now with our production department. 

Kind regards, 

on behalf of

Dr. Fernanda Penido Matozinhos 

Academic Editor

PLOS ONE